# Echocardiographic Characterization of Left Heart Morphology and Function in Highly Trained Male Judo Athletes

**DOI:** 10.3390/ijerph19148842

**Published:** 2022-07-21

**Authors:** Jelena Slankamenac, Aleksandra Milovancev, Aleksandar Klasnja, Tamara Gavrilovic, Damir Sekulic, Marijana Geets Kesic, Tatjana Trivic, Violeta Kolarov, Patrik Drid

**Affiliations:** 1Faculty of Sport and Physical Education, University of Novi Sad, 21000 Novi Sad, Serbia; jelly.95@live.com (J.S.); ttrivic@yahoo.com (T.T.); 2Faculty of Medicine, University of Novi Sad, 21000 Novi Sad, Serbia; aleksandra.milovancev@mf.uns.ac.rs (A.M.); aleksandar.klasnja@mf.uns.ac.rs (A.K.); violeta.kolarov@mf.uns.ac.rs (V.K.); 3Institute of Cardiovascular Diseases of Vojvodina, 21204 Sremska Kamenica, Serbia; 4Serbian Institute of Sport and Sports Medicine, 11000 Belgrade, Serbia; tamara.gavrilovic@rzsport.gov.rs; 5Faculty of Kinesiology, University of Split, 21000 Split, Croatia; dado@kifst.hr (D.S.); markes@kifst.hr (M.G.K.); 6Institute for Pulmonary Diseases of Vojvodina, 21204 Sremska Kamenica, Serbia

**Keywords:** athlete’s heart, left ventricular hypertrophy, ventricular remodeling, cardiac adaptation, heart geometry

## Abstract

The long-term practice of judo can lead to various changes in the heart including increased dimensions of the left ventricle in diastole and thickening of the interventricular septum and the posterior wall of the left ventricle. This study aimed to assess left ventricular morphology and function in elite male judokas. A comparative cross-sectional study was conducted that included a total of 20 subjects, 10 judokas, and 10 healthy non-athletes aged 24 ± 2.85 years. Demographic and anthropometric data were analyzed. All subjects underwent a medical examination and a two-dimensional transthoracic echocardiogram. Different parameters of left ventricular morphology and function were measured and compared between athletes and non-athletes. Left ventricle mass and LV mass index were higher in judokas than in non-athletes (*p* < 0.05), as well as PW thickness (9.78 ± 0.89 mm vs. 8.95 ± 0.76 mm). A total of six (*n* = 6) of athletes had eccentric hypertrophy, while others had normal heart geometry. LVEDd, LVEDs, LVEDd/BSA, and LVEDs/BSA were significantly higher in judokas (*p* < 0.05). LVEDd in athletes ranged from 48 to 62 mm. These values, combined with normal diastolic function, ejection fraction, and shortening fraction, indicate that the judokas’ cardiac adaptation was physiological rather than pathological.

## 1. Introduction

A regular training program for elite athletes often causes morphological changes in the heart and blood vessels, increases the thickness of the left ventricular (LV) wall (hypertrophy) and ventricular mass, causes LV cavity enlargement (dilatation), and enhances resting parasympathetic tone [1]. The changes entail the functional, structural, and electrical remodeling of the heart that accompanies regular training [2]. These changes represent a characteristic clinical picture of the “athlete’s heart”, an important physiological adaptation that helps athletes better perform physical tasks than non-athletes [3].

Different types of activity create different forms of cardiac hypertrophy [4]. Dynamic exercise mainly leads to an increased work frequency and heart rate, increasing the minute volume. Vascular resistance in the systemic circulation decreases, which leads to slightly lower blood pressure. Increased load and stretching of the myocardium affect the increase in the internal dimensions of the left ventricle and a proportional thickening of the LV wall, which causes eccentric hypertrophy. Static exercise leads to an increase in heart rate and blood pressure. These hemodynamic changes cause concentric LV hypertrophy. Concentric hypertrophy leads to the thickening of the LV wall but not to an increase in its internal dimensions [4,5]. Absolute cardiac dimensions depend on several factors, such as gender, body size, sporting discipline, weekly training hours, ethnicity, drugs, and genetics. For this reason, to assess the geometry of the left ventricle, it is necessary to use the left ventricular mass index and relative wall thickness (RWT) in distinguishing physiological from pathological hypertrophy, like hypertrophic cardiomyopathy [6,7,8]. Indexing parameters are significant because body size can be responsible for about 50% of the variability of the cardiac structural changes [9]. Intense exercise causes cardiovascular changes that are sometimes associated with significant risk. Although rare in young athletes, hypertrophic cardiomyopathy is the leading cause of sudden cardiac death [10]. Recent studies show that most male athletes have altered echocardiographic parameters relative to reference values. These changes are more pronounced in athletes who engage in high-intensity endurance sports. On the other hand, female athletes generally show normal echocardiographic parameters, and any deviation from the reference values in women should be further investigated [11,12,13].

To be successful in any sport, athletes should achieve a high level of physical conditioning and physical fitness. Judokas show great anaerobic capacity and strength when exercises involve the upper body. Judo, as a high-intensity intermittent sport, relies mainly on anaerobic sources, although other sources also contribute significantly to performance. The aerobic component in judoka contributes to better and faster recovery during short breaks between efforts [14].

Judo is a dynamic, high-intensity, intermittent sport [15]. Practicing judo can lead to various changes in the heart. Long-term practice of this sport can lead to increased dimensions of the left ventricle in diastole and the thickening of the interventricular septum and the posterior wall of the left ventricle. Changes in the heart morphology in judokas are more similar to those that occur in endurance athletes than in strength athletes [16,17]. The most important reason for studying this problem is the need to differentiate the physiological state of the heart from the pathological one. We had the opportunity to conduct research on elite Olympic-level judokas, whereas most previous studies were conducted on athletes competing at the national level. The aim of the study was to examine the left ventricular structure and function of elite male judokas.

## 2. Materials and Methods

At the Serbian Institute for Sports and Sports Medicine and the Institute of Cardiovascular Diseases of Vojvodina, a comparative cross-sectional study was conducted that included a total of 20 subjects divided into 2 groups. The first group consisted of 10 male judokas. The second group consisted of 10 healthy non-athletes, all male subjects. All participants were Caucasians. The sample of judokas was purposive and included the best judokas from Serbia who participated in more than five international competitions that are scored for the Olympic norm. The athletes had 24 h of high-intensity training per week. All judokas had started with judo training at the age of seven. The second group was randomized from non-sporting groups. Demographic, anthropometric, and echocardiographic data were analyzed for all subjects. The average age of the judokas was 25.5 ± 3.17 years and the non-athletes’ group average was 22.5 ± 1.43 years. All participants underwent a medical examination and 2D transthoracic echocardiogram. Exclusion criteria were: a previous history of hypertension, heart diseases, hepatic, kidney, malignant and infections disorders, diabetes mellitus, and electrolyte imbalance.

The analysis of anthropometric characteristics included the measurement of body height (BH), which was performed with an anthropometer according to Martin (GPM, Switzerland), with an accuracy of 0.1 cm, in a standing position with heels together, toes apart, and hands close to the body. The position of the head during the measurement of body height was horizontal, and the values are expressed in cm. Body mass (BM) was determined using an Omron weight scale BF511 (Omron, Osaka, Japan) with an accuracy of 0.1 kg. The body surface area (BSA) was calculated according to the appropriate formula:$$BSA\;\left(m^{2}\right)=\frac{\sqrt{\mathrm{BH}\;\left(\mathrm{cm}\right)}\times \;\mathrm{BM}\;\left(\mathrm{kg}\right)\;}{3600}$$

### 2.1. Echocardiography

All subjects underwent two-dimensional transthoracic echocardiography using a low-frequency probe of 2.5 MHz on an ultrasound device, Vivid 9, produced by General Electric, Boston, MA, USA. For each acquisition, three cardiac cycles of uncompressed data were stored in a cine-loop format and estimated by one researcher who was blinded to the clinical characteristics of the subjects. The examination was performed in the supine position (left lateral decubitus), and the measurements and calculations of morphological and functional indicators of the left ventricle were according to the standards of the American Association for Echocardiography and the European Association for Cardiovascular Imaging [18].

The following parameters were measured and calculated: The parasternal long-axis view was used to measure the thickness of the posterior wall of the left ventricle in diastole (PWd, cm), the thickness of the interventricular septum in diastole (IVSd, cm), the left ventricular dimensions (end-systolic diameter, LVIDs; end-diastolic diameter, LVIDd; aortic root; separation of cusps; and ascending aorta dimensions expressed in mm). The left ventricular myocardial hypertrophy level was calculated based on the formula LVHL = (IVSd + PWd)/2. The relative LV wall thickness (RWT) was calculated from the formula RWT = 2 × PWd/LVIDd. Left ventricular mass (LVM) was automatically calculated via area–length in the software according to measures obtained from the echosonographic parameters acquired from the parasternal cross-sectional view in which mid-ventricular diastolic and systolic epicardial and endocardial surfaces were observed without papillary muscles; diastolic and systolic mitral-to-apical distance in the apical 4C view was also acquired: LVM = 1.05 × (0.8 × A1 × (L + t)) − (0.8 × A2 × L). Specific muscle gravity is 1.05 g/mL; A1 is end-diastole epicardial area (cm); A2 is end-diastole endocardial area (cm); L is end-diastole ventricular length; and t is average wall thickness (cm). LV mass index was calculated by indexing with the patient’s body surface area (BSA m^2^). Left ventricular ejection fraction (EF%) was calculated according to the modified Simpson disk method EF = ((EDVLV − ESVLV)/EDVLV) × 100%, and left ventricular shortening fraction (FS) was expressed in %.

LV geometry is defined based on RWT and LVM index. Left ventricular morphology is classified into one of four groups: normal geometry—RWT < 0.42 and LVMI ≤ 115 g/m^2^; concentric remodeling—RWT ≥ 0.42 and LVMI ≤ 115 g/m^2^; concentric hypertrophy—RWT ≥ 0.42 and LVMI >115 g/m^2^; or eccentric hypertrophy—RWT < 0.42 and LVMI > 115 g/m^2^.

### 2.2. Diastolic Function

LV diastolic function was assessed based on the speed of mitral annulus movement by tissue Doppler with the determination of the following parameters: Early wave E (cm/sec) begins with the opening of the mitral valve, and atrial wave A (cm/sec) indicates the rate of late atrial filling of the left ventricle associated with atrial contraction. E/A ratio and the septal and lateral velocity of early movement of the mitral ring, “e’sep” and “e’lat,” were calculated; the ratio of early transmitral inflow and the mean early mitral ring velocity (E/e’mean) were also calculated.

### 2.3. Statistics Methods

The data obtained in this way were sorted in Microsoft Excel (version 2019, Microsoft Inc., Redmond, Washington, DC, USA), while the statistical data processing used the software package Statistical Package for Social Sciences (SPSS, version 20.0. IBM Corp., Armonk, NY, USA). Data processing included descriptive and inferential statistics. Numerical features are presented as means, measures of variability (range of values, standard deviation), and attributive features using frequencies and percentages. Depending on the nature of the data, the comparison of the values of numerical features between the two groups was performed using Student’s t-test, while the differences in the frequencies of attribute features were compared using the chi-square test. The results are presented in tables and graphs. The significance level was set at *p* < 0.05.

## 3. Results

The average age of the judokas was 25.5 ± 3.17 years, and the non-athletes’ average age was 22.5 ± 1.43 years. Table 1 shows the body height, body weight, and body surface area of the participants. There were no differences between these parameters (*p* > 0.05).

On comparison of the left ventricular hypertrophy level and relative wall thickness between the athletes and the non-athletes, no statistical significance was obtained. Relative wall thickness was normal in all subjects (<0.42). There was a difference between groups in left ventricle mass in that the absolute values were higher in the judokas (233.44 ± 68.75 g). There was also a difference in LV mass indexed to the body surface area (*p* <0.05). In the control group, seven (*n* = 7) subjects had normal heart geometry, while the others (*n* = 3) had concentric LV remodeling. A total of six (*n* = 6) athletes showed eccentric hypertrophy, while the others (*n* = 4) had normal heart geometry (Figure 1).

Comparing the echocardiographic characteristics of the examined groups (Table 2), the absolute internal dimensions of the left ventricular cavity and the end-diastolic and end-systolic diameters were significantly higher in the athletes compared with the non-athletes (*p* < 0.05). The average end-diastolic diameter of the left ventricle in the judokas was 55.2 ± 5 mm, ranging from 48 mm to 62 mm, while in the non-athletes, it was 47.00 ± 3.26 mm, ranging from 43 mm to 52 mm (*p* < 0.05). The end-systolic diameters were 35.2 ± 5.5 mm in the judokas and 30.90 ± 3.90 mm (*p* < 0.05) in the non-athletes. The average LVEDd/BSA in athletes was 27.22 mm/m^2^, while in the control group, it was significantly lower, 23.89 mm/m^2^ (*p* < 0.05). There was a significant difference between the groups concerning LVEDs/BSA (*p* < 0.05), and athletes showed higher average values (17.36 ± 1.5 mm/m^2^ vs. 15.65 ± 1.1 mm/m^2^). The thickness of the posterior wall in athletes was, on average, 9.78 mm, while in non-athletes, it was significantly lower, 8.95 mm (*p* < 0.05). There was no statistically significant difference in interventricular septal thickness.

The absolute aortic root dimensions and cusp separation values differ statistically significantly between the judokas and the control group (*p* < 0.05). However, there were no significant differences in the ascending aortic dimensions.

Systolic function was preserved in all subjects and did not differ significantly between groups. The judokas’ ejection fraction was 63.10 ± 2.99%, while the values in the second group were slightly higher, 64.70 ± 1.49%, although statistically insignificant.

Early E-wave (cm/s) did not differ between athletes and non-athletes. There was a statistically significant difference in the late atrial filling (A), as well as in the E/A ratio. The judokas and the control group also differed significantly in e’sep. However, when we indexed E with e’sep and e’av, there was no statistically significant difference. E/e’av was within normal limits, ≤8 in both the athlete and non-athlete groups.

## 4. Discussion

This study shows that in our small sample, highly trained judokas exhibited significant changes in heart morphology and function compared with non-athletes. The internal dimensions of the heart, LVEDd, and LVEDs were significantly higher in the judokas compared with the control group. The difference is retained when these values are indexed with the body surface area. The judokas also had significantly increased left ventricle posterior wall thickness and left ventricular mass. Most of them (*n* = 6) had eccentric left ventricle hypertrophy, while the others had normal heart geometry, which may be related to endurance training combined with strength training. Although there were differences in systolic and diastolic heart function between the athletes and non-athletes, these parameters were within normal limits in both groups.

Judo is a dynamic sport that combines the effects of isokinetic and isometric exercise, for which reason judokas can have LV wall thickness values > 13 mm [19]. In our study, all of the athletes had a wall thickness < 11 mm. Whyte et al. [20] reported that the normal upper limit of wall thickness observed in 442 athletes was 14 mm, although 11 subjects had values greater than 14 mm. In Sun’s study [21], only 3/339 athletes had LV wall thickness values greater than 13 mm. Sports that combine resistance and endurance training are a strong stimulus for left ventricular wall thickening, and normal upper limits can be 16 mm [22].

Left ventricle hypertrophy level and relative wall thickness were similar between athletes and non-athletes. There was no significant difference. In the literature, we find that most athletes have normal heart geometry, but abnormal geometry is not rare and depends on the type of sport, race, and training time [23]. Urhausen et al. [19] reported that the upper limit of LVM/BSA was 170 g/m^2^, while in our study, the average LVMI was 105.16 ± 24.89 g/m^2^. The upper normal LV mass index, according to Lang, is 115 g/m^2^ [18]. The size of our study sample was small (*n* = 20), so future research should involve more athletes so that we can draw more definite conclusions.

Markedly increased dimensions of the heart cavity can be found in athletes in response to long-term intensive and systematic training, especially in those who engage in aerobic sports, which entail a physiological adaptation of the heart [24]. Dilated cardiomyopathy is a primary heart muscle disease characterized by the enlargement of the left ventricular cavity, systolic dysfunction, and normal left ventricular wall thickness [25]. It is important to distinguish between athletes’ physiological heart characteristics and structural heart disease. Athletes with DCM are at increased risk of sudden cardiac death, and in most cases, they are excluded from competitiosn [26]. Pelliccia et al. [24] examined the internal dimensions of the heart in 1309 athletes engaged in various sports. Thirteen judokas took part in the research. The mean LVEDd in the judokas was 56.2 ± 5.2 mm. Most subjects had LVEDd within the generally accepted normal range, ≤54 mm [27,28], while 31% had LVED ≥ 60 mm. Although 20% (*n* = 2) of our judokas had LVED ≥ 60 mm, none of the athletes showed signs of global systolic dysfunction or abnormal diastolic filling, suggesting that ventricular enlargement was more likely due to intense sports training rather than a consequence of primary heart disease [29].

Elite athletes have a larger aortic root compared with control groups, especially in sinus Valsalva. The enlargement of the aortic root is greater in the Valsalva sinus than in the aortic valve annulus because it is fibrous and less extensible. Aortic root dimensions ranged from 30.2 to 33.1 mm [30]. These results indicate that a slight increase in the dimensions of the aorta in athletes is a normal adaptation to training. A substantial increase in aortic root size implies a pathological process. In our study, athletes had significantly larger aortic root dimensions compared with sex-matched non-athletes. The dimensions of the ascending aorta did not differ significantly between groups. According to Lang’s recommendations for cardiac chamber quantification by echocardiography in adults, aortic root dimensions in normal adults are on average 34 mm, and the dimension of the proximal part of the ascending aorta are 30 mm [18].

EF and FS did not differ between the judokas and the control group, and the values of these parameters were within normal limits. Previous research also shows that EF is preserved in athletes, indicating that resting contractility is not increased [31,32].

The early wave (E) was similar in judokas and non-athletes. However, the A-wave was significantly higher in the control group. Considering the difference in the A-wave velocity, there is also a difference in the E/A ratio, which was significantly higher among the judokas. Despite this difference, there is no abnormality in this ratio between LV filling and relaxation. For all participants, the E/A ratio was >1, which is similar to other studies [33,34]. Caselli found that despite the increase in filling during early diastole, the E-wave velocity was not increased in athletes [35]. This is most likely due to relatively extended diastole and an enlarged LV cavity. The speed of the early movement of the mitral valve, e’, did not differ between our athletes and healthy non-athletes. Absolute values of this parameter were normal in all participants. Values of e’ > 8 cm/s are normal for elite athletes, while e’ < 8 cm/s should be interpreted carefully [33,36]. An average E/e’ ≤ 8 combined with normal LV filling pressure excludes LV diastolic dysfunction.

A common finding in athlete’s heart is an increased left ventricular mass associated with thickening of the walls and an enlarged inner diameter of the left ventricle [37]. The normal upper limits of left ventricular wall thickness depend on several factors, such as gender, age, race, BSA, genetics, and sports.

Our study also provides insight into the dimensions of the heart to define the upper limits of left ventricular dimensions associated with long-term and intense training of judokas, such as athlete’s heart. The recognition of the upper limits of left ventricular size in athletes has clinical significance and offers the opportunity to distinguish benign heart remodeling, seen with increased physical exertion, from pathological structural changes such as dilated and hypertrophic cardiomyopathy [24,38,39]. Distinguishing physiological increases in heart dimensions from pathological increases is the basis for reducing the risk of sudden cardiac death in athletes and avoiding unnecessary disqualification from sports in the case of a false positive HCM [40].

The study has several limitations. First, participants in our study were highly trained judokas, Caucasians, and <30 years old. For these reasons, the results may not be applicable to recreational athletes, athletes of different ethnic backgrounds, and ages. Second, due to the cross-sectional nature of the study, no follow-up was conducted, which limits the understanding of the nature of the remodeling process. Third, we used BSA to index cardiac dimensions, which is sometimes not the most acceptable method for estimating LV size in athletes. Other studies suggest using LV dimensions normalized per height, lean mass, and allometric scaling [41,42] because scaling LV structural data for differences in body surface area within subjects may influence the interpretation of the results [31]. Finally, our sample size was small; only ten top-level judokas participated in the research. Future research should be extended to a larger number of respondents. Nuclear magnetic resonance is the gold standard for determining the morphology of the heart and coronary heart disease. For this reason, in future research, MRI should be used as a non-invasive radiological method of determining cardiac structure in athletes [43].

## 5. Conclusions

In the present study, highly trained male judo players demonstrated significantly higher LV posterior wall thickness and increased left ventricular mass compared with sex-matched non-athletes. The internal dimensions of the left ventricle were also significantly increased in judokas. Although most athletes had eccentric left ventricular hypertrophy, RWT and LVmass index were within normal limits. These changes in left heart morphology, combined with normal diastolic function, ejection fraction, and shortening fraction, indicate that the judokas’ cardiac adaptation was physiological rather than pathological, showing the pattern of left ventricular remodeling commonly observed in strength and power training, with a limited endurance training contribution. This research provides insight into the upper limits of heart dimensions in judokas and can help clinicians distinguish between athlete’s heart and primary heart disease, especially HCM and DCM.

Future research should be expanded to a larger number of subjects and should also include female judokas. Our study included the best judo players from Serbia, who participated in a large number of international competitions and achieved Olympic norm, which distinguishes our research from most published research.

## Figures and Tables

**Figure 1 ijerph-19-08842-f001:**
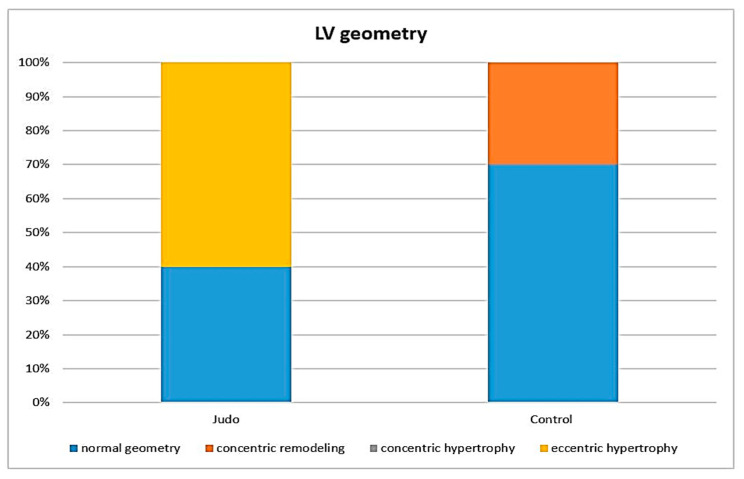
The left ventricular geometry in the judokas and the controls.

**Table 1 ijerph-19-08842-t001:** The age and anthropometric parameters (BH, BM, BSA) of the judokas and the control group.

Variable	Judo X ± SD	Non-Athletes X ± SD	*p*
AGE (years)	25.50 ± 3.17	22.50 ± 1.43	0.014
BH (cm)	181.90 ± 8.17	180.40 ± 6.23	0.650
BM (kg)	82.11 ± 10.43	77.80 ± 13.70	0.439
BSA (m^2^)	2.02 ± 0.17	1.97 ± 0.17	0.463

BH-body height; BM-body mass; BSA-body surface area; *p*-statistical significance; SD-standard deviation.

**Table 2 ijerph-19-08842-t002:** Echocardiographic characteristics of the study population.

	Judo (*n* = 10) X ± SD	Non-Athletes (*n* = 10) X ± SD	t	*p*
**LV geometry**				
LVHL	9.78 ± 0.95	9.12 ± 0.61	1.842	0.082
RWT (mm)	0.36 ± 0.02	0.38 ± 0.04	−1.789	0.090
LVM (g)	233.44 ± 68.75	137.70 ± 16.20	4.286	0.000
LVMI (g/m^2^)	105.16 ± 24.89	69.92 ± 6.45	4.334	0.000
**Heart cavity dimensions**				
LVEDd (mm)	55.20 ± 5.05	47.00 ± 3.26	4.311	0.000
LVEDd/BSA (mm/m^2^)	27.22 ± 1.34	23.89 ± 0.93	6.424	0.000
LVEDs (mm)	35.20 ± 4.49	30.90 ± 3.90	2.286	0.035
LVEDs/BSA (mm/m^2^)	17.36 ±1.53	15.65 ± 1.16	2.812	0.012
IVS (mm)	9.84 ± 0.11	9.30 ± 0.67	1.350	0.194
PW (mm)	9.78 ± 0.89	8.95 ± 0.76	2.232	0.039
**Aortic dimensions**				
AORTIC ROOT (mm)	29.10 ± 4.86	22.90 ± 2.13	3.692	0.002
CUSPIS SEPARATION (mm)	22.30 ± 2.16	19.30 ± 0.67	4.187	0.001
ASCENDING AORTA (mm)	27.40 ± 1.58	27.00 ± 1.15	0.647	0.526
**Systolic function**				
EF%	63.10 ± 2.99	64.70 ± 1.49	−1.510	0.148
FS%	36.30 ± 3.58	34.39 ± 5.15	0.963	0.348
**Diastolic function**				
E–wave (cm/s)	0.82 ± 0.07	0.91 ± 0.14	−1.824	0.085
A (cm/s)	0.43 ± 0.05	0.56 ± 0.09	−3.978	0.001
E/A	1.91 ± 0.18	1.62 ± 0.17	3.643	0.002
e’sep (m/s)	0.13 ± 0.02	0.15 ± 0.02	−2.967	0.008
e’lat (m/s)	0.18 ± 0.03	0.18 ± 0.02	−0.185	0.855
e’av (m/s)	0.16 ± 0.02	0.17 ± 0.02	−1.791	0.090
E/e’av	5.36 ± 0.69	5.49 ± 1.02	−0.317	0.755

A—late atrial contraction; E-wave—early wave; EF—ejection fraction; e’av—average peak early velocity; E/e’ av—early wave to average peak early velocity; e’ lat—lateral peak early velocity; e’ sep—septal peak early velocity; FS—fraction of shortening; IVS—interventricular septum; LVEDd—LV end-diastolic diameter; LVEDd/BSA—ratio of LV end-diastolic diameter and body surface area; LVEDs—LV end-systolic diameter; LVEDs/BSA—ratio of LV end-systolic diameter to body surface area; LVHL—left ventricle hypertrophy level; LVM—left ventricle mass; LVMI—left ventricle mass index; p—statistical significance; PW—posterior wall; RWT—relative wall thickness; SD—standard deviation.

## Data Availability

The data presented in this study are available on request from the corresponding author.

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
