# Peer review of "Echocardiographic Characterization of Left Heart Morphology and Function in Highly Trained Male Judo Athletes"

_ijerph, 2022, doi:10.3390/ijerph19148842_

Round 1

Reviewer 1 Report

The authors submit a novel paper detailing the cross-sectional findings of heart morphology and function in a small sample of elite Judokas and non-trained, health adults. The preliminary findings indicate that there are structural changes present in the majority of their elite sample, however do not appear to be pathological or disadvantageous, as cardiac function is preserved. This small study provides preliminary data to understand the effects of prolonged, elite level Judo training - involving both resistive and aerobic capacity - on cardiac adaptations and will help to inform future, larger trials to accurately define normal parameters in these athletes to better understand the presence of any non-training related cardiac abnormalities.

I have a few comments below.

Abstract:

1. Background: Practicing judo can lead to various changes in the heart. Long-term 17 practice of this sport can lead to increased dimensions of the left ventricle in diastole, thickening of 18 the interventricular septum, and the posterior wall of the left ventricle

Please change to: 'Background: Long-term practice of Judo can lead to various changes in the heart including increased dimensions of the left ventricle in diastole, and thickening of the interventricular septum and the posterior wall of the left ventricle'

Introduction

2. Although rare in young athletes, hypertrophic cardiomyopathy is the leading cause of sudden cardiac death - please add reference.

3. Please comment on  the novelty of this study in the final part of your introduction to further justify the need for this study.  I.e. is it the first, or different to current literature - has it been done before?, etc. 

Methods

4. Please provide descriptive statistics of the ethnicity and training history of Judokas and comparators to aid with interpretation of data.

5. Line 104 - 113: The examination was performed in the supine position (left  lateral decubitus), and measurements and calculations of morphological and functional indicators of the left ventricle according to the standards of the American Association for  Echocardiography and the European Association for Cardiovascular Imaging [16]. The following parameters were measured and calculated: Parasternal long–axis view  was used to measure the thickness of the posterior wall of the left ventricle in diastole (PWd, cm), the thickness of the interventricular septum in diastole (IVSd, cm), left ventricular dimensions: end-systolic (LVIDs) and end-diastolic (LVIDd) diameter, aortic root, separation of cusps, and ascending aorta dimensions (expressed in mm). The left ventricular myocardial hypertrophy level was calculated based on the formula: LVHL = (IVSd + PWd) / 2. The relative LV wall thickness (RWT) was calculated from the formula: RWT=2xPLWd/LVIDd.

I can see from the referenced guidelines where these techniques are derived - the equations seem to be relatively simple diameter calculations derived from the computed dimensions of each component. However, these specific equations are not listed in the referred document. If these equations are common knowledge, or have a references to justify, can you please add or further clarify in the text. Additionally, please clarify the difference between PWd and PLWd, the latter of which does not seem to have been defined. 

6. By comparing the values of the left ventricular hypertrophy level and relative wall 162 thickness in athletes and the control group, no statistical significance was obtained. 163 Relative wall thickness was normal in all subjects (<0.42). There was a difference between 164 groups in values of the left ventricle mass. The absolute values are higher in judokas 165 (233.44 ± 68.75 g). Also, there is a difference in LV mass indexed to the body surface area 166 (p <0.05). In the control group, 70% (n=7) of subjects had normal heart geometry, while 167 30% (n=3) had concentric LV remodeling. A total of 60% of athletes showed eccentric 168 hypertrophy, while others (n=4) had normal heart geometry (Figure 1).

I think given the small sample size it is not appropriate to use % values (i.e. 60%) as this could be misleading.  Instead it is more appropriate to just report n (i.e n=6). Please fix throughout. 

7. In the discussion, please start each paragraph with a topic sentence to introduce the comparisons made, rather than beginning with 'X et al states...'. Please fix in the multiple paragraphs where this is the case.

8. More caution needs to be applied in the statements made in the discussion based off the small sample that the analysis is based upon. For instance, please refrain from making definitive statements and qualify each by acknowledging that these relationships were seen in the sample. For example:

'This study shows that highly trained judokas exhibit significant changes in heart 210 morphology and function compared to non-athletes. '

'This study shows that in our small sample, highly trained judokas exhibit significant changes in heart morphology and function compared to non-athletes.'

9. Conclusion - please provide more information regarding the future direction of investigation - i.e. larger samples, diversity, better imaging techniques, etc. Also, please refer back to the novelty of this study - is it the first, or different to current literature. 

I wish you good luck with your submission and look forward to seeing future research in this area building upon these interesting findings.

Author Response

 Abstract:

  1. Background: Practicing judo can lead to various changes in the heart. Long-term 17 practice of this sport can lead to increased dimensions of the left ventricle in diastole, thickening of 18 the interventricular septum, and the posterior wall of the left ventricle

Please change to: 'Background: Long-term practice of Judo can lead to various changes in the heart including increased dimensions of the left ventricle in diastole, and thickening of the interventricular septum and the posterior wall of the left ventricle'

Thank you for the suggestion, we have corrected it.

Introduction

  1. Although rare in young athletes, hypertrophic cardiomyopathy is the leading cause of sudden cardiac death - please add reference.

Thank you for pointing out the omission, we have added the reference.

  1. Please comment on  the novelty of this study in the final part of your introduction to further justify the need for this study.  I.e. is it the first, or different to current literature - has it been done before?, etc. 

We had the opportunity to conduct research on elite Olympic-level judokas, whereas most previous studies were conducted on athletes competing at the national level.

Methods

  1. Please provide descriptive statistics of the ethnicity and training history of Judokas and comparators to aid with interpretation of data.

All participants were male Caucasians. We included the best judokas from Serbia who participated in more than five international competitions that are scored for the Olympic norm. The athletes had 24 hours of high-intensity training per week. All judokas have started with judo training at the age of seven.

  1. Line 104 - 113: The examination was performed in the supine position (left  lateral decubitus), and measurements and calculations of morphological and functional indicators of the left ventricle according to the standards of the American Association for  Echocardiography and the European Association for Cardiovascular Imaging [16]. The following parameters were measured and calculated: Parasternal long–axis view  was used to measure the thickness of the posterior wall of the left ventricle in diastole (PWd, cm), the thickness of the interventricular septum in diastole (IVSd, cm), left ventricular dimensions: end-systolic (LVIDs) and end-diastolic (LVIDd) diameter, aortic root, separation of cusps, and ascending aorta dimensions (expressed in mm). The left ventricular myocardial hypertrophy level was calculated based on the formula: LVHL = (IVSd + PWd) / 2. The relative LV wall thickness (RWT) was calculated from the formula: RWT=2xPLWd/LVIDd.

I can see from the referenced guidelines where these techniques are derived - the equations seem to be relatively simple diameter calculations derived from the computed dimensions of each component. However, these specific equations are not listed in the referred document. If these equations are common knowledge, or have a references to justify, can you please add or further clarify in the text. Additionally, please clarify the difference between PWd and PLWd, the latter of which does not seem to have been defined. 

  1. LVHL formula is simple and commonly used formula for assessing left ventricular hypertrophy level.
  2. RWT has reference, formula is listed in the referenced document (Lang, R.M.; Badano. L.P.; Mor-Avi, V.;Afilalo, J.; Armstrong, A.;Ernande, L. et al. Recommendations for cardiac chamber quantification by echocardiography in adults: an update from the American Society of Echocardiography and the European Association of Cardiovascular Imaging. Heart J. Cardiovasc. Imaging 2015, 16(3), 233–271.)

On the 1445 page.

PWd and PLWd is the abbreviation for the same parameter, it is corrected in the text. 

  1. By comparing the values of the left ventricular hypertrophy level and relative wall 162 thickness in athletes and the control group, no statistical significance was obtained. 163 Relative wall thickness was normal in all subjects (<0.42). There was a difference between 164 groups in values of the left ventricle mass. The absolute values are higher in judokas 165 (233.44 ± 68.75 g). Also, there is a difference in LV mass indexed to the body surface area 166 (p <0.05). In the control group, 70% (n=7) of subjects had normal heart geometry, while 167 30% (n=3) had concentric LV remodeling. A total of 60% of athletes showed eccentric 168 hypertrophy, while others (n=4) had normal heart geometry (Figure 1).

I think given the small sample size it is not appropriate to use % values (i.e. 60%) as this could be misleading.  Instead it is more appropriate to just report n (i.e n=6). Please fix throughout. 

Thank you for the suggestion, we have corrected it throughout the text.

  1. In the discussion, please start each paragraph with a topic sentence to introduce the comparisons made, rather than beginning with 'X et al states...'. Please fix in the multiple paragraphs where this is the case.

Thank you for the suggestion, we have corrected it in all paragraphs.

  1. More caution needs to be applied in the statements made in the discussion based off the small sample that the analysis is based upon. For instance, please refrain from making definitive statements and qualify each by acknowledging that these relationships were seen in the sample. For example:

'This study shows that highly trained judokas exhibit significant changes in heart 210 morphology and function compared to non-athletes. '

'This study shows that in our small sample, highly trained judokas exhibit significant changes in heart morphology and function compared to non-athletes.'

Thank you for the suggestion, we have corrected it.

  1. Conclusion - please provide more information regarding the future direction of investigation - i.e. larger samples, diversity, better imaging techniques, etc. Also, please refer back to the novelty of this study - is it the first, or different to current literature. 

Future research should be extended to a larger number of subjects, and should also include female judokas. Nuclear magnetic resonance is the gold standard for determining the morphology of the heart and coronary heart disease, and for this reason, in future research, MRI should be used as a non-invasive method to determine cardiac structure. Our study included the best judo players from Serbia, who participated in a large number of international competitions and thereby achieved the Olympic norm, which differs from most published research.

Reviewer 2 Report

I am grateful for the opportunity to review the interesting topic „Echocardiographic Characterization of Left Heart Morphology and Function in Highly Trained Judo Male Athletes“.

The manuscript is an interesting one. However, I have some concerns regarding the further applicability of the results obtained.

1. The summary must be adjusted to the requirements of the IJERPH.

2. L 60-61: There seems to be a lack of reference to literature.

3. L 66-70: The text of paper namely Introduction as well as Materials and Methods should be complemented by data on the physical characteristics of judo athletes (in terms of aerobic/anaerobic capacity).

4. L 76:  What type of cross-sectional study has been carried out? Was this a comparative cross‑sectional study?

5. L 74: I propose that the Materials and Methods section be supplemented by information on the sampling procedure.

6. L 78: “control group” must be changed throughout the text of the manuscript. There can't be a control group in a cross-sectional study. The methodology does not allow it. consists“ could be changed to consisted“.

7. L 81: Table 1 displays the results, therefore this information must be described in the Results section.

8. L 101, 134: Please provide ISO (International Organization for Standardization) numbers for these devices.

9. L 94: “(?2)” must be changed to (?2). The same problem is observed on lines 127, 130.

10. I propose to improve the resolution of Figure 1.

11. While writing, it is necessary to use past tenses throughout the entire text of this paper.

12. The Discussion Unit cannot support the results along with values (numbers, e.g. on lines 213-216); thus I suggest that the rewriting of results could be used more moderately.

13. This manuscript of the Authors is submitted to the IJERPH. Therefore, the Conclusions must identify exceptional practical implications for a specific population group too; thus practical implications seem to have to be written by the Authors.

King Regards

Author Response

  1. The summary must be adjusted to the requirements of the IJERPH.

Thank you for the suggestion, we have corrected it.

  1. L 60-61: There seems to be a lack of reference to literature.

Thank you for the suggestion, we have corrected it.

  1. L 66-70: The text of paper namely Introduction as well as Materials and Methods should be complemented by data on the physical characteristics of judo athletes (in terms of aerobic/anaerobic capacity).

Thank you for the suggestion, we added it in Introduction.

  1. L 76:  What type of cross-sectional study has been carried out? Was this a comparative cross‑sectional study?

It was comparative cross‑sectional study.

  1. L 74: I propose that the Materials and Methods section be supplemented by information on the sampling procedure.

The sample of judokas was purposive and included the best judokas from Serbia who participated in more than five international competitions that are scored for the Olympic norm. The athletes had 24 hours of high-intensity training per week. All judokas have started with judo training at the age of seven. The second group was randomized from the non-sporting groups.

  1. L 78: “control group” must be changed throughout the text of the manuscript. There can't be a control group in a cross-sectional study. The methodology does not allow it. “consists “could be changed to “consisted“.

Thank you for the suggestion, we have corrected it.

  1. L 81: Table 1 displays the results, therefore this information must be described in the Results section.

Thank you for the suggestion, we described it in Results section.

  1. L 101, 134: Please provide ISO (International Organization for Standardization) numbers for these devices.

Martin anthropometer (GPM, Switzerland).

Omron weight scale BF511 (Omron, Japan).

  1. L 94: “(?2)” must be changed to (?2). The same problem is observed on lines 127, 130.

Thank you for the suggestion, we have corrected it.

  1. I propose to improve the resolution of Figure 1.

Thank you for the suggestion, we have corrected it.

  1. While writing, it is necessary to use past tenses throughout the entire text of this paper.

Thank you for the suggestion, we have corrected it.

  1. The Discussion Unit cannot support the results along with values (numbers, e.g. on lines 213-216); thus I suggest that the rewriting of results could be used more moderately.

Thank you for the suggestion, we have corrected it.

  1. This manuscript of the Authors is submitted to the IJERPH. Therefore, the Conclusions must identify exceptional practical implications for a specific population group too; thus practical implications seem to have to be written by the Authors.

This research provides insight into the upper limits of heart dimensions in top-level judokas and can help clinicians distinguish between athlete's hearts and primary heart disease, especially HCM and DCM in athletes.